# From Gut Microbiota through Low-Grade Inflammation to Obesity: Key Players and Potential Targets

**DOI:** 10.3390/nu14102103

**Published:** 2022-05-18

**Authors:** Claudia Vetrani, Andrea Di Nisio, Stavroula A. Paschou, Luigi Barrea, Giovanna Muscogiuri, Chiara Graziadio, Silvia Savastano, Annamaria Colao

**Affiliations:** 1Department of Clinical Medicine and Surgery, Endocrinology Unit, University of Naples “Federico II”, 80131 Naples, Italy; c.vetrani@libero.it (C.V.); chiaragraziadio@live.it (C.G.); metabolismounina@gmail.com (S.S.); operafederico2@gmail.com (A.C.); 2Department of Medicine, Operative Unit of Andrology and Medicine of Human Reproduction, University of Padova, 35128 Padova, Italy; andrea.dinisio@gmail.com; 3Endocrine Unit and Diabetes Centre, Department of Clinical Therapeutics, Alexandra Hospital, School of Medicine, National and Kapodistrian University of Athens, 11527 Athens, Greece; s.a.paschou@gmail.com; 4Dipartimento di Scienze Umanistiche, Università Telematica Pegaso, 80143 Napoli, Italy; luigi.barrea@unina.it; 5Centro Italiano per la Cura e il Benessere del Paziente con Obesità (C.I.B.O), University of Naples “Federico II”, 80131 Naples, Italy; 6UNESCO Chair “Education for Health and Sustainable Development”, University of Naples “Federico II”, 80131 Naples, Italy

**Keywords:** gut microbiota, obesity, low-grade inflammation, gut–brain axis, fat–gut axis, leaky gut, intestinal permeability, microbial metabolites

## Abstract

During the last decades, the gut microbiota has gained much interest in relation to human health. Mounting evidence has shown a strict association between gut microbiota and obesity and its related diseases. Inflammation has been appointed as the driving force behind this association. Therefore, a better understanding of the mechanisms by which gut microbiota might influence inflammation in the host could pave for the identification of effective strategies to reduce inflammation-related diseases, such as obesity and obesity-related diseases. For this purpose, we carried out an extensive literature search for studies published in the English language during the last 10 years. Most relevant studies were used to provide a comprehensive view of all aspects related to the association of gut microbiota and low-grade inflammation with obesity. Accordingly, this narrative review reports the evidence on the key players supporting the role of gut microbiota in the modulation of inflammation in relation to obesity and its complications. Moreover, therapeutic approaches to reduce microbiota-related inflammation are discussed to provide potential targets for future research.

## 1. Introduction

In recent years, the scientific interest in mechanisms underlying obesity and its related diseases has increased. Specifically, the gut microbiota has gained significant attention as a central target for the investigation of metabolic disorders [1]. It is widely known that the human gut hosts thousands of different bacterial species, which underwent a coevolutionary process with the human host for millions of years through a mutual relationship with the host [2].

Gut microbiota consists of around 100 trillion organisms, thereby including bacteria, viruses, *fungi*, and bacteriophages [3,4]. Gut microbiota could be divided into three families: eukaryotes-containing genome-separating nuclear membrane from cellular material, bacteria, and archaea, which relates to prokaryotes deprived DNA-containing nucleus. Dominant microbial *phyla* in the human gut are *Bacteroidetes* and *Firmicutes*. Indeed, they make up more than 90% of all bacteria; some common gut bacterial genera are *Bacteroides*, *Fusobacterium*, *Peptococcus*, *Escherichia*, *Clostridium*, *Peptostreptococcus*, *Lactobacillus*, *Eubacterium*, *Ruminococcus*, and *Bifidobacterium* [3,4,5,6]. 

The physiological functions of the gut microbiota range from metabolic regulation, food ingredients digestion, synthesis of new elements, intestinal barrier protection by synthesis of mucine and epithelial regeneration, immune response by regulating the intestinal immune system, thus affecting the human host metabolism at multiple levels [5]. On these bases, the alteration of gut microbiota composition, a condition known as “gut dysbiosis”, can be involved in the pathogenesis of various disorders such as obesity and its related diseases [7].

More in detail, gut dysbiosis is characterized by the presence of bacteria with increased activity in energy harvesting from the diet. Indeed, studies in mice demonstrated that increased *Firmicutes* abundance is related to a greater fat deposition despite the lower chow intake, likely through the reduction of energy expenditure. This suggests that the gut microbiota is a key regulator of body fat storage [8,9]. In addition, as recently reviewed [10,11], the gut microbiota might influence the endocrine signals to the hypothalamus (both orexigenic and anorexigenic hormones), thus influencing food intake. Overall, this might lead to an imbalance between energy intake and energy expenditure. Consequently, unburned calories are stored in adipocytes, thus leading to increased body fat. 

Accumulation of adipose tissue is strongly associated with reduced oxygen availability and cellular suffering, which induce the activation of the immune system and the overflow of inflammatory cytokines [12]. Therefore, excess body weight—as occurs in obesity—is a trigger for the development of low-grade inflammation. 

Chronic inflammation is a risk factor for increased morbidity and mortality for several diseases, e.g., cardiovascular disease, cancer, diabetes, chronic kidney disease, non-alcoholic fatty liver disease, and autoimmune and neurodegenerative disorders [13,14]. 

Although inflammation, particularly in the visceral adipose tissue, was appointed as a major cause of obesity-associated comorbidities [13,14], microbiota might play a key role also in regulating intestinal immunity and, therefore, local and systemic inflammation associated with obesity and its related diseases. 

Against this background, the present narrative review aims to highlight the different mechanisms explaining the association between gut microbiota and obesity-related low-grade inflammation. Moreover, potential strategies to manage this association are discussed. 

## 2. Mechanisms Explaining the Association between Gut Microbiota and Obesity-Related Low-Grade Inflammation

### 2.1. Microbiota Composition

Although the specific composition of gut microbiota greatly differs depending on age, sex, and ethnicity [15], several observational and experimental studies have highlighted common features in the microbial community of mice [16] and individuals with obesity [17,18]. Overall, evidence showed an association between obesity and *Firmicutes* abundance. In addition, lower diversity of the gut microbiota is associated with increased levels of circulating pro-inflammatory cytokines and chemokines. Interestingly, individuals with reduced diversity were “non-responders” to weight loss intervention, with no improvement in markers of systemic inflammation, insulin resistance, and dyslipidemia [17]. These findings suggested a causal link between gut microbiota modifications, obesity, and inflammation.

As for the specific microbial composition, some studies comparing obese vs. lean individuals reported a higher abundance of bacteria from *Firmicutes phylum* (i.e., *Blautia hydrogenotorophica*, *Coprococcus catus*, *Eubacterium ventriosum*, *Ruminococcus bromii*, and *Ruminococcus obeum*) in people with obesity whereas normal-weight individuals presented more bacteria from *Bacteroidetes phylum* (i.e., *Bacteroides faecichinchillae and Bacteroides thetaiotaomicron*), leading to a higher *Firmicutes/Bacteroidetes* ratio) [18,19]. 

Interestingly, when genetically obese mice (ob/ob, leptin-deficient) were studied, it was revealed that an obesity-associated microbiota pattern can provide higher amounts of energy from certain ingested foods to the host compared with a lean-associated gut microbiota [20,21]. Moreover, an increased abundance of pro-inflammatory bacteria (i.e., *Escherichia coli*) with the consequent reduction of bacteria with anti-inflammatory properties (i.e., *Fecalibacterium prausnitzii*) has been observed in people with obesity [22,23]. Nevertheless, to the best of our knowledge, no studies identified specific microbial species to be used as potential targets to treat obesity and inflammation.

### 2.2. Microbial Metabolites

Two different mechanisms have been proposed to explain the role of gut dysbiosis in obesity-related inflammation: the release of bacterial components (lipopolysaccharide, LPS) and microbial metabolites (short-chain fatty acids, SCFA) [24]. Indeed, the gut microbiota synthesizes a wide range of metabolites, such as nitric oxide (NO), gamma-aminobutyric acid (GABA), SCFA, and catecholamines [25]. These metabolites are recognized by host receptors in different target tissues, resulting in direct modulation of energy metabolism and host endocrine regulation. As an example, microbial catecholamines can modify nutrients’ absorption in the gut, whereas SCFA can modify the activity of G protein-coupled receptor (GPR) 41 and 43, thus reducing the activity of lipoprotein lipase with consequent accumulation of triglycerides in the adipose tissue [26]. Additionally, GABA, one of the most important inhibitory neurotransmitters in the central nervous system, can be produced by a particular set of bacteria from the genus *Lactobacillus*, with consequent modification of food intake regulation at the brain level and subsequent body weight modification [27]. In addition, the metabolites produced by gut microbiota can also exert anti- or pro-inflammatory effects by stimulating macrophage activity [28]. Several studies focused on LPS, the components of the outer membrane of Gram-negative bacteria that are released into circulation when bacteria have been neutralized. Notably, the direct administration of LPS can induce systemic inflammation and insulin resistance in animal models [29]. In humans, several studies have shown an increase in LPS and LPS-binding protein (LBP) in serum from patients with obesity, metabolic syndrome, or type 2 diabetes [30,31,32]. Moreover, macrophages can detect LPS through the TLR4 receptor, and, as a consequence, they switch from M2 to M1 phenotype. This phenomenon is linked to increased secretion of pro-inflammatory cytokines (i.e., IL-1 and TNF-α) [28]. Conversely, SCFA (mainly butyrate) have been shown to mitigate the LPS-induced accumulation of macrophages, suggesting a potential anti-inflammatory effect in obesity [33]. On the other hand, subjects undergoing weight loss interventions by bariatric surgery showed an improvement in glucose tolerance and a significant reduction of both LPS and LBP concentrations [34,35]. 

Overall, these data support the role of microbial metabolites in the modulation of inflammatory status in the host.

### 2.3. Intestinal Permeability

The intestinal barrier is a large structural feature covering 400 m^2^ and consumes approximately 40% of the energy of the body. The most important functions of the intestinal barrier are the absorption of nutrients from ingested food and the prevention of the entry of antigens and toxins from the gut lumen into the blood. Moreover, the barrier contributes to the maturation of body immunity and helps immune cells to differentiate the self-antigens from non-self-antigens [36,37]. 

The maintenance of intestinal barrier integrity is strictly dependent on tight junctions, which represent the interface between adjacent epithelial cells, selectively allowing the passage of molecules across the barrier. Some important proteins produced by tight junctions are members of the zona-occludens family (ZO-1, 2, 3), claudins, adhesion molecules, and occludins [37,38]. Therefore, any perturbation of tight junctions’ function or stability can lead to the impairment of intestinal barrier integrity and consequent increase in its permeability. 

The impairment of intestinal permeability allows for the leakage of bacterial metabolites or, in general, bacterial antigens from the intestinal lumen into the circulation, thus inducing an immune response in the host with the promotion of a pro-inflammatory state. This condition, named “leaky gut”, has been consistently reported as a causative agent for various metabolic diseases such as obesity, insulin resistance, and inflammation-related diseases [38,39]. The driving force behind this association is chronic low-grade inflammation in the body due to continuous leakage of antigens from the gut lumen to the systemic circulation [38,39].

Several studies have reported a crucial role of gut microbiota in the maintenance of intestinal barrier integrity due to the aforementioned interactions, mainly with the immune system of the host. This important regulatory function is achieved by the synthesis of important elements by intestinal bacteria, such as vitamins, antigens, and SCFA. Interestingly, SCFA (mainly butyrate) are released by gut microbiota using undigested carbohydrates and serve as an energy source for colonocytes [40,41].

In addition, butyrate has been shown to improve the integrity of the intestinal barrier by up-regulating ZO-1 and claudin-1 expression, which is involved in the formation of the intestinal barrier’s tight junctions. The resulting increase in barrier integrity leads to a significant reduction in the leaking of LPS, therefore reducing insulin resistance and overall dysmetabolic function, as reported in animal models [42,43,44]. Similarly, in women, the presence of bacteria from the genus *Faecalibacteria* led to a reduction of intestinal permeability and inflammation thanks to butyrate production by these bacterial species [45]. 

As for the association of bacterial species, *Akkermansia* spp. populations have been shown to regulate gut permeability by increasing tight-junctions function, which leads to weight loss and reduced glucose tolerance in mice on a high-fat diet [46]. In addition, in obese mice, lower abundance of Bifidobacterium induced the downregulation of GLP-2 synthesis, a key protein involved in the regulation of gut barrier function. As a result, the integrity of the intestinal epithelial barrier was compromised in these mice, thus increasing gut permeability [47]. 

### 2.4. Gut–Brain–Fat Axis 

Energy balance is regulated by a complex neuroendocrine network that connects the gut, brain, and fat. This continuous communication between the gastrointestinal tract, brain, and adipose tissue affects appetite, food intake, energy consumption, and energy storage [48,49,50]. After meal ingestion in the gastrointestinal tract, various signals are sent towards the brain and fat to inform about the changes in the nutritional status. Gut microbiota plays a pivotal role in the function of the gut–brain–fat axis, as they convert dietary nutrients into metabolites that convey, in turn, the appropriate messages both centrally and peripherally. These microbiota-derived molecules affect the adipose tissue but also modulate the hypothalamus and brain, resulting in crucial ultimate modifications of the host metabolism [48,51]. 

As reported above, gut microbiota can also release LPS, which can, in turn, activate the immune response, inflammasome, and cytokine production [51]. More in detail, the induction and differentiated maturation of B cells, T cells, or dendritic cells can also lead to various grades of systemic inflammation, which affects adipose cells size and phenotype, ultimately increasing insulin resistance [51]. These changes affect various central and peripheral signals and the function of the gut–brain–fat axis in total, possibly inducing dysregulation of the host physiology and the activation of a pro-inflammatory milieu that can cause neuroinflammation [48]. These effects might also explain why people with obesity are more prone to develop neurodegenerative diseases linked to inflammation (Alzheimer’s disease and Parkinson’s disease) [52]. 

The described signals can change in a diet-dependent manner, but microbiota can also differentiate the response to a certain diet [48,49,50,51]. The biochemical and metagenomic analysis provided evidence that this happens via differentially increased action of fermentation enzymes and nutrient transporters, which result in various metabolites and therefore various signals to fat and the brain [53]. 

On the other hand, some studies showed that gut endocrine cells may be influenced by the gut microbiota through the expression of bacterial recognition receptors, such as Toll-like receptors, or receptors for microbial metabolites, such as SCFA. This can activate pro-inflammatory or anti-inflammatory pathways, thus demonstrating the role of gut endocrine cells in mucosal defense [52]. Moreover, microbial metabolites have been shown to stimulate the vagal nerve, thus inducing an indirect anti-inflammatory reflex by activating the sympathetic nervous system [54]. Therefore, microbial metabolites are key players in the modulation of the gut–brain and gut–brain–fat axis, which can be used as potential targets to reduce inflammation.

### 2.5. Gut Microbiota and Inflammasome/Cytokines Interplay

The inflammasome is an innate immune signaling complex. This complex is formed by a variety of cells when the inflammasome receptors or sensors recognize signals known as pathogen-associated molecular patterns (PAMPs) or danger-associated molecular patterns (DAMPs) [51,55].

Inflammasome activation is associated with rapid innate inflammatory response and leads mainly to the production of IL-1β and IL-18 [55,56]. The inflammation process that is triggered by the inflammasome contributes to the pathogenesis of a wide spectrum of chronic diseases, including obesity [51,57]. 

Gut microbiota releases metabolites, such as LPS or other PAMPs or DAMPs, that activate the immune response, inflammasome, and cytokine production. Latest developments have suggested that this route is bidirectional, as inflammasome can also regulate gut microbiota composition and gut–brain–fat axis. Indeed, on some occasions, inflammasome results in the production of certain antimicrobial peptides in intestinal epithelial cells that modulate the gut microbiota profile [51]. 

Differences in diet can affect gut microbiota profile and, in turn, the activity of the inflammasome. For example, a ketogenic or low-calorie diet that increases the production of the ketone body β-hydroxybutyrate ultimately contributes to the inhibition of the NLR family pyrin domain-containing 3 (NLRP3) inflammasome [58]. In a similar way, a diet rich in omega-3 fatty acids inhibits activation of the NLRP3 inflammasome by signals of the metabolite-sensing receptors GPR40 and GPR120 (also known as FFAR4). The inhibition of inflammation prevents, in these cases, diet-induced insulin resistance [59]. On the other hand, a diet rich in fibers contains short-chain fatty acids that bind to the metabolite-sensing receptor GPR43 of enterocytes and activate the NLRP3 inflammasome [60].

Conversely, a diet rich in cholesterol or saturated fatty acids has been shown to promote activation of the NLRP3 inflammasome and to increase insulin resistance in mice [61,62]. Such studies have provided evidence that dietary nutrients affect gut microbiota, which then modulates inflammasome activity, cytokine production, and susceptibility to the development of metabolic diseases [48,51].

Mice that have genetic alterations on NLRP3 or IL-18 present a gut microbiota profile that is associated with inflammation, exacerbated hepatic steatosis, and obesity. Interestingly, these phenotypes are transmissible to wild-type mice with the transfer of gut microbiota [63]. Mice with alterations in sensors of inflammasome are prone to hepatosteatosis and obesity after a high-fat diet [51,63]. Such abnormalities were treated with the use of antibiotics, suggesting a sound connection of gut microbiota with inflammasome and obesity [63]. An experimental study investigated mice lacking NLRP3. These mice were fed a high-fat and high-carbohydrate diet compared with other mice fed with standard chow. It was shown that there were differences in gut microbiota dysbiosis between the two groups, particularly in the *Prevotellacee* abundance. The change in dysbiosis was associated with increased concentrations of triglycerides in the liver and feces as well as with intestinal permeability, adipose tissue inflammation, and liver injury [64]. When wild-type mice were fed a high-fat, high-carbohydrate diet compared with standard chow, the degree of dysbiosis was not so substantial, indicating the essential role of both diet and inflammasome in regulating gut microbiota diversity and metabolic diseases. Another interesting experimental study showed that a diet high in fat and cholesterol for mice carrying a mutation predisposing them to spontaneous inflammatory bone disease resulted in the relative abundance of *Prevotella* microbiota type. This was associated, in turn, with reduced expression of pro-IL-1β in the mice intestine [65].

All the above data, even mainly experimental, imply that the interplay between inflammasome and gut microbiota axis can represent a centerpiece in the development of obesity and other inflammatory clinical conditions. Multiple immune mechanisms are operated in both intestinal epithelial cells and immune cells. This connection has to do with the ability of gut microbiota and inflammasome to regulate one another [51]. Of course, the causative microbial population responsible for specific inflammatory responses and clinical manifestations, such as obesity, remains to be elucidated.

## 3. Therapeutic Opportunities

Modulation of gut microbiota has been speculated to be a novel tool to improve low-grade inflammation that leads to obesity and its metabolic complications. 

Dietary changes represent the first strategy to improve the composition of microbial communities. Indeed, viscous and readily fermented fibers (e.g., β-glucan and arabinoxylans from whole-grain, pectins from fruit, vegetables, and legumes, and resistant starch) have been shown to increase the abundance of butyrate-producing bacteria, which may positively influence the inflammatory status of the host [66,67]. Notably, a higher abundance of butyrate-producing bacteria has been associated with better outcomes during weight-loss interventions [68,69].

Moreover, prebiotics (inulin, berberine, polyphenols) and probiotics (*Akkermansia muciniphila*, *Lactobacillus reuteri*, *Faecalibacterium prauznitsii*) have been suggested as potential strategies to reduce inflammation, mostly in animal models [67]. However, to date, no conclusive evidence is available in humans due to methodological limitations of the studies (small sample size, short duration, and heterogenic treatment groups). 

More recently, some authors reported new insights on the effect of anti-obesity drugs on microbiota composition. Indeed, liraglutide has been shown to reduce body weight by influencing the composition of the gut microbiota in simple obese and diabetic obese rats [70,71]. Notably, in vitro and animal studies showed that liraglutide can modulate inflammatory gene expression in peripheral blood mononuclear cells and intestinal immune cells [71,72]. As for other anti-obesity drugs, a recent study [73] investigated the effects of 42-day treatment with four drugs (sibutramine, tacrolimus/FK506, bupropion, and naltrexone), alone or in combination in obese rats. Sibutramine showed a greater effect on weight loss and reduction of the expression of flagellum-encoding genes, which have been previously associated with inflammation. In addition, despite a smaller effect on body weight changes, bupropion and naltrexone significantly increase the abundance of members of the Bacteroidetes *phylum*. 

Nevertheless, to the best of our knowledge, the link between anti-obesity drugs, microbiota, and inflammation has not yet been demonstrated in humans. 

## 4. Conclusions and Future Perspectives

This narrative review aimed to highlight all aspects explaining the association between gut microbiota, inflammation, and obesity. Indeed, the literature on this topic is complex and heterogeneous, and it might not be suitable for systematic reviews. However, due to the inclusion criteria (relevant papers published in the last 10 years), we can not exclude that we missed some information. 

Overall, the main findings suggested that gut microbiota plays a pivotal role as a contributor to systemic inflammation associated with obesity and its complications. 

The triggers for this inflammatory state have not been fully identified, but recent evidence supports the role of microbial metabolites in several pathways (Figure 1).

Overall, data available so far show increased gut permeability with consequent passive diffusion of bacterial antigens leading to chronic inflammation. In particular, LPS is the main contributor to the onset of pro-inflammatory pathways, whereas SCFA released by gut microbiota plays a protective role in the maintenance of intestinal barrier integrity, thus preventing chronic inflammation. Inflammation can induce the impairment of several functions in the host (e.g., insulin sensitivity, gut hormone secretion, and regulation of the gut–brain axis). Therefore, chronic inflammation with the consequent excess fat storage is the trigger for obesity and a detrimental “vicious cycle”. 

Nevertheless, evidence on the causal role of microbiota composition and activity in regulating inflammation comes mainly from reviews. In addition, original articles retrieved with our literature search were mainly studies in animal models, as summarized in Table 1. 

Therefore, it is necessary to translate emerging experimental evidence into clinical results to highlight reliable therapeutic targets.

Dysbiosis-related inflammation and production of detrimental microbial metabolites can induce the impairement of several functions leading to obesity (Figure 2). 

Among the mechanisms, the increased energy harvest in the intestine, the impaired regulation of food intake, and inflammation in adipose tissue significantly contribute to excess fat storage and, therefore, to obesity and its complications.

Nevertheless, it is important to underline that current evidence does not allow to exclude that obesity and associated lifestyle factors might increase low-grade inflammation, and these together affect the gut microbiota. Despite this, the identification of effective strategies to modulate microbiota composition and reduce dysbiosis will improve low-grade inflammation related to obesity and its complications, with relevant beneficial effects on public health.

## Figures and Tables

**Figure 1 nutrients-14-02103-f001:**
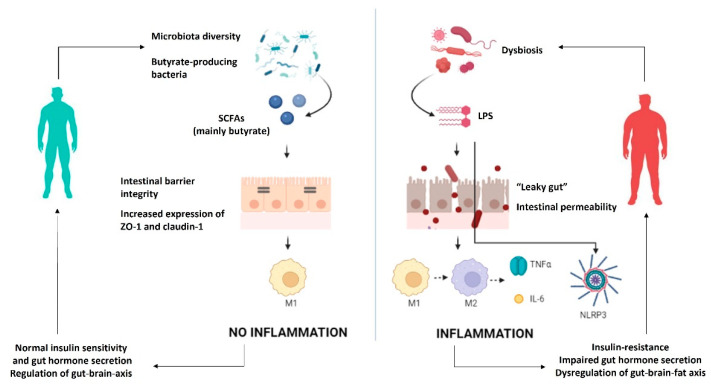
Mechanisms by which gut microbiota can modulate low-grade inflammation and obesity. IL-6—interleukin 6; LPS—lipopolysaccharides; M1—macrophages M1; M2—macrophages M2; NLRP3—inflammasome; SCFA—short-chain fatty acids; TNFα—tumor necrosis factor α; ZO-1—zonulin.

**Figure 2 nutrients-14-02103-f002:**
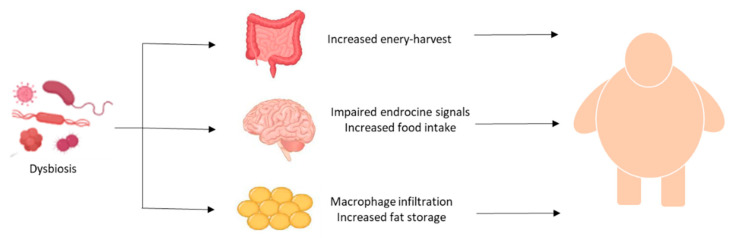
Main mechanisms linking dysbiosis and obesity.

**Table 1 nutrients-14-02103-t001:** Summary table of studies focusing on the association between microbiota composition and inflammation/obesity.

Main Outcome	Type of Study	Main Findings
Microbiota composition	Studies in mice	- Increased *Bacteroidetes* abundance in lean mice [16] - Bacteria with increased energy-harvesting activity [20,21]
	Human studies	- Reduced diversity in overweight and obese individuals is associated with high hs-CRP levels [17] - Increased *Firmicutes* abundance in overweight and obese individuals [17,18,19] - Increased abundance of pro-inflammatory bacteria (e.g., *Escherichia coli*) while reduced bacteria with anti-inflammatory properties (e.g., *Fecalibacterium prausnitzii*) [22,23]
Microbial metabolites	In vitro studies	- Butyrate reduces LPS concentration and activation of macrophages [33]
	Studies in mice	- *Lactobacillus* increase the production of GABA, reducing food intake [27] - LPS infusion increases TNF-α levels [29]
	Human studies	- LPS concentration is associated with obesity, MS, and T2D [30,31,32,33] - Weight loss (bariatric surgery) decrease LPS concentrations [34,35]
Intestinal permeability	Studies in mice	- Butyrate increase the expression of mucin and tight junction proteins [43] - *Akkermansia* in lean mice increase thigh junction function [46]
	Human studies	- *Faecalibacteria* abundance is associated with reduced zonulin levels in overweight women [45]
Gut microbiota and inflammasome/cytokines interplay	Studies in mice	- Dysbiosis increases inflammation and obesity [63] - *Prevotella* abundance reduces inflammation [64,65]

GABA—γ-aminobutyric acid; hs-CRP—high sensitivity C-reactive protein; LPS—lipopolysaccharide; MS—metabolic syndrome; T2D—type 2 diabetes.

## Data Availability

Not applicable.

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
