# Peer review of "From Gut Microbiota through Low-Grade Inflammation to Obesity: Key Players and Potential Targets"

_nutrients, 2022, doi:10.3390/nu14102103_

Round 1
Reviewer 1 Report
In the present study (nutrients-1680842), Vetrani et al., reviewed the association of Gut microbiota and low-grade inflammation with obesity. The topic is exciting and in the limelight in the current scenario of obesity research.
Besides this, the present manuscript lacks novelty and doesn't contribute much to the scientific interests in this particular area of research. A simple search could render an ample number of reviews covering all aspects of this paper, while the present manuscript superficial covers some of the areas. A systematic search and discussion of the findings from preclinical and clinical studies (separately) could have made this paper more comprehensive. The best possible way of enhancing the quality and novelty of this manuscript may be to convert it into a systematic or scoping review by following a standard methodology.
My comments are as follows:
- Title- the phrase "a dangerous journey" is hard to digest.
- Abstract- It should also provide some information on the methods used for literature search.
- The introduction is very brief. It should provide more details on the background and the relationship between obesity, GMB, and inflammation.
- Method- This section is not so important for the narrative reviews. It makes no sense that studies were excluded based on the impact factor quartile of the journals. In this way, the authors must have eliminated several important pieces of information.
- Systematic search needs to be robust using different databases, not only PubMed but OVID databases such as Medline, Scopus, etc. Further, there should be inclusion and exclusion criteria for the studies during screening. The article should be included/excluded not on the basis journal's impact factor but based on the set criteria.
- In the other sections, there should be a clear indication of the origin of the information (preclinical or clinical), and each section could be subdivided accordingly.
- The inclusion of tables showing the association of GMB-Inflammation-Obesity from clinical and preclinical studies and their qualitative representation could have made this article more comprehensive.
- In section 4, line 277, citation #61 doesn't seem appropriate.
- Liraglutide is not the only anti-obesity drug that alters gut microbiota. Again, the systematic search could have provided more extensive information.
- The conclusion section needs a little more expansion
- Figure 1 shows half of the information i.e., gut dysbiosis-inflammation; no correlation is made with obesity.
Reviewer 2 Report
In the manuscript titled "From gut microbiota through low-grade inflammation to obesity: a dangerous journey" Claudia Vetrani and colleagues, they have reported that a better understanding of the mechanisms by which gut microbiota might influence inflammation in the host could pave for the identification of effective strategies to reduce inflammatory-related diseases, such as obesity and obesity-related diseases. I have a few comments regarding the present review.
-Maybe a better introduction of how adipose tissue and gut microbiota could be related is necessary. In addition, new information about why obesity is associated with low-grade inflammation and which are the health implications is necessary.
-How was the quality of the studies evaluated, more than the impact factor, the authors have added other tools of differentiation?
-A table summarizing the most important information maybe is required.
-Reading the review I have in my mind, was is new that this review gives to the literature, what is the novelty or this?
Round 2
Reviewer 1 Report
Dear Authors,
I appreciate your efforts in improving the manuscript and addressing my earlier comments. However, the majority of the suggestions are not taken seriously. My further comments are as follows:
- The title looks improved.
- Remove the phrase "carried out a scoping review with" from the abstract since this paper doesn't comply with the guidelines of a scoping review.
- The introduction looks good to me now. Please try to merge the last 4 paragraphs into one.
- The present form of the manuscript doesn't comply with the structure and guidelines of the scoping review.
- It is highly recommended to remove the method section since it remains a narrative review.
- The authors say they have conducted an extensive search, but no additional findings are reported; however, the inclusion of table 1 sounds good.
- NOTE- Scoping review has- defined inclusion and exclusion criteria; systematic search; screening of title/abstract and full text; and qualitative representation of findings in the form of tables or graphs, which is completely missing from the study.
- Furthermore, articles can not be excluded based on time duration of publication (for ex., last 10 years) and impact factor quartiles. It should be selected or rejected based on inclusion/exclusion criteria.
- In response to the previous comment #11, others added figure 2 but the original question remained as unanswered. Authors are advised to please refer to the previous comment (11. Figure 1 shows half of the information i.e., gut dysbiosis-inflammation; no correlation is made with obesity.), and modify the figure 1 accordingly.
Reviewer 2 Report
Thank you to the authors for taking into account my previous comments, no further comments from my side, the manuscript look great now.
Author Response
We thank the reviewer for helping us to improve the manuscript.